# Muscle Changes during Direct Attack under Different Conditions in Elite Wheelchair Fencing

**DOI:** 10.3390/sports12070188

**Published:** 2024-07-10

**Authors:** Julio Martín-Ruiz, Jorge Alarcón-Jiménez, Nieves de Bernardo, Ignacio Tamarit-Grancha, Xavier Iglesias, Laura Ruiz-Sanchis

**Affiliations:** 1Department of Health and Functional Assessment, Catholic University of Valencia, 46900 Valencia, Spain; julio.martin@ucv.es; 2Department of Physiotherapy, School of Medicine and Health Sciences, Valencia Catholic University, 46900 Valencia, Spain; jorge.alarcon@ucv.es (J.A.-J.); nieves.debernardo@ucv.es (N.d.B.); 3Department of Physical Preparation and Conditioning, Catholic University of Valencia, 46900 Valencia, Spain; ignacio.tamarit@ucv.es; 4Institut Nacional d’Educació Física de Catalunya, Universidad de Barcelona, 08038 Barcelona, Spain; xiglesias@gencat.cat; 5Department of Sports Management and Physical Activity, Catholic University of Valencia, 46900 Valencia, Spain

**Keywords:** electromyography, wheelchair fencing, direct attack

## Abstract

Wheelchair fencing is a medium-distance combat sport in which the wheelchair is fixed to the floor. This requires a maximum concentration and gestational speed. Refined techniques and fatigue tolerance are essential to achieve competitive results. Aim: The main objective of this study was to measure the speed and muscular participation of a direct attack gesture with and without fatigue in a sample of elite wheelchair fencers. Methods: The maximal isometric force, gestural speed, and resistance to gestural velocity were estimated in 10 elite performed direct fencers. Results: The results revealed that sitting height and wingspan were important factors in achieving the highest speed, especially in women (r = 0.9; *p* = 0.07). Other factors, such as the elbow angle where the closed position was better, affected muscle contraction in both categories of athletes (*p* = 0.01). The onset of fatigue was earlier in category B than in category A, with greater variation in direct attack movement. The results highlight the importance of analyzing direct attacks for the best application of quick force, speed, and muscle participation, as well as identifying the onset of technical deterioration to devise a competitive strategy. These parameters may allow for precise design of conditioning sessions for elite wheelchair shooters.

## 1. Introduction

Wheelchair fencing (WCF) is one of the oldest sports practiced by athletes with disabilities [1], and has been performed since the first Paralympic Games (Rome, 1960). It is an offshoot of conventional fencing with three weapons (foil, épée, and sabre), and follows similar tactics and rules [2,3]. To participate in WCF, one or more of the following physical disabilities must be present: hypertonia, ataxia, athetosis, limb deficiency, muscle weakness, and short range of motion [4]. 

The International Wheelchair and Amputee Sports Federation [4] has defined three categories of athletes. Category A athletes are the most functional and demonstrate good seated balance and fencing arm function (e.g., amputees or individuals with spinal cord injuries below T10). Those in Category B demonstrated correct balance in the sitting position and good arm function (e.g., paraplegic fencers with spinal cord injuries below T10 or at T9–10). Finally, the fencers in Category C demonstrated neither good balance when seated nor functionality of the fencing arm (e.g., quadriplegics with spinal cord injuries at C5–C8). Research on WCF has been limited to exploring the physiological demands of sports [5,6], injury epidemiology [3], and kinematic and electromyographic analyses of thrust [2,7,8]. Although these studies provide valuable information, they are not sufficient to understand the physical attributes that underpin performance. 

Appropriate execution of movements in a wheelchair is important for obtaining optimum results and injury prevention. Borysiuk et al. demonstrated that patients with poor trunk muscle control had a higher prevalence of injuries (4.9/1000 h) than those with adequate control (3.0/1000 h) [9]. The authors highlighted that the elbow (32.6%) and shoulder (15.8%) were commonly affected, which is supported by the results of Fairbairn et al. [10]. Therefore, trunk stabilization exercises may prevent such injuries. Overall, upper limb, trunk, neck, and head injuries accounted for 26.4%, 21.4%, and 5% of all injuries, respectively, with significant differences between men and women [11]. Other areas, such as the hands and feet, are also affected [12]. Although WCF has a lower physiological demand than seated fencing [13], it is more physically demanding. Therefore, paying close attention to the technical aspects of shoulder girdle movement is recommended to prevent injury [3]. 

Fencing is a combat acyclic sport in which critical speed is an important parameter, either in force application or resistance to force velocity, throughout a series of movements. These actions require coordination to develop suitable techniques [14,15]. In addition, the anticipation and discrimination of actions within the shortest possible time are important requirements [16]. Arm speed is a key parameter for favorable outcomes. Milic et al. [17] indicated that, to achieve the highest efficiency, the fencer should always perform actions as quickly and accurately as possible. Villiere et al. [18] added strength, power, flexibility, and trunk and arm motor control to sports requirements. The average time required by a fencer to perform a defensive action is 0.353 ± 0.028 s [19]; therefore, an attacking maneuver within this threshold duration is the target. In Category A fencers, the role of the unarmed arm is appreciated by coaches for its positive impact, although not all fencers have such an advantage, which prevents its generalization to Category B and C athletes [18]. 

Competitor experience is crucial for achieving good reaction speed [20]. Reaction time increases with the number of stimuli and is lower in novices even during fatigue. However, the decision-making interval is shorter for experienced fencers [21,22]. To assess the segmental speed in wheelchairs, experiments were conducted in which the shooter responded to a visual stimulus. These experiments highlight the use of surface electromyography (EMG) for such assessments. Borysiuk et al. [2] measured the activation in Category A and B fencers. The former generates activation in the torso and lumbar muscles, followed by the acromial deltoid (AD), oblique abdomen, extensor carpi longus (EL), triceps, and biceps. In Category B fencers, the first stimulation was noted in the EL, followed by the external oblique, latissimus dorsi, biceps, deltoids, and triceps muscles. Therefore, differences in muscle activation between the groups were observed. In another study, Group A fencers demonstrated a shorter contraction time than Group B shooters, thus concluding that antagonist activation was better in Group A fencers [23]. These studies highlight the technical differences between groups of fencers without providing a unified set of required technical gestures, especially those involving trunk intervention. Additionally, these studies focused on visual non-free conditioned gestures and were unable to generate the maximum speed. Therefore, conditioning the muscles according to their actions is necessary, making isometric studies useful in this context. In contrast, it allows an increase in the maximum activation of each muscle group [24] and accurately identifies the proportion of participation of each group of muscles in a gesture because each sport and athlete has different strength timings and peaks [25]. This makes it possible to program stimuli objectively without resorting to non-specific standardized loads. 

Focusing on the duration of a fencing competition, fatigue has a negative influence owing to several inherent technical nuances [26] such as short and rapid actions with little recovery time. An important gap in the literature is the fatigue of gestural speed following frequent exposure, for example, in a fencing competition characterized by intermittent actions of maximum intensity. Therefore, in this study, we aimed to estimate the maximal and dynamic isometric muscle activation along with the timing of a direct attack, considering the shoulder and arm muscles, which would allow the creation of a comparative model of various categories of athletes in this sport. As evidence demonstrates that the reaction times and speeds are higher under competitive conditions than under non-competitive conditions, the results of these tests may be useful references for further research [27]. Maximal and dynamic isometric muscle activation, as well as the action timing of a direct attack, do not depend on the weapon used, as a fencer will perform similarly to all three weapons, which is associated with fencing skills rather than the type of weapon used [28].

However, analyzing fatigue following repeated actions by analyzing muscle changes will help to determine individual time thresholds and improve conditional training. Consequently, the strategy can be determined individually in this wheelchair sport without the need to refer to the Olympic fencing. Additionally, we aimed to provide new methods for the assessment of arm movements in fencing, which will enable the assessment of neuromuscular fatigue, as such tests in combat sports are lacking [29].

## 2. Materials and Methods

### 2.1. Experimental Approach to the Problem

The following three tests were designed ad hoc to determine the force variables: (1) a test of maximal isometric force in the direct attack action for 5 s using EMG and dynamometry; (2) a test of direct attack at maximum speed using EMG and the initial bending angle; and (3) a fatigue test with 20 direct attacks at maximum speed using EMG. The tests were conducted in the morning, between 10 a.m. and 1 p.m. The evaluated muscles included the AD, clavicular deltoid (CD), triceps brachii (TB), and extensor longus (EL). The data were analyzed to evaluate any association between the variables of muscle activation and direct attack to determine whether athletes with better coactivation during a direct attack had a shorter lapse and less fatigue while performing the action repeatedly, irrespective of sex and practice category. 

A pilot test was performed on wheelchair-fencing athletes from provincial clubs to validate the protocol and its subsequent application. The duration of the tests included 15 min of warm-up and 20 min for the three tests.

### 2.2. Participants 

The study included 10 WCF athletes (six and four athletes from Categories A and B, respectively). These international athletes are ranked worldwide and belong to the Spanish national wheel fencing team. Participants with less than 5 years of training and/or those with any injury that prevented safe testing were excluded. Table 1 summarizes the basic anthropometric data and elbow angles with respect to the time lapse of direct attacks in these fencers. The tests were explained to the athletes and informed consent was obtained from all participants. This study was approved by the Research Ethics Committee (CEI) of the Catholic University of Valencia (Approval No. UCV/2022-2023/107). This study was performed in accordance with the Declaration of Helsinki for Human Clinical Trials.

### 2.3. Procedure 

The sitting height and wingspan were evaluated for each participant by an ISAK-accredited team member. Weight and standing height were excluded because they could not perform these assessments across the study sample. For sitting height, the participant was seated on a 44 cm high bench. A stadiometer (Seca 213; Seca, Hamburg, Germany) was placed on the vertex in an upright position and the head was positioned until the Frankfort angle was reached. The bench height was subtracted from the bench height to obtain the sitting height. For the wingspan, the participants performed shoulder abduction and elbow extension with the phalanges extended while sitting on a stool without a backrest and with both hips and shoulder blades in contact with the wall. The maximum distance between the middle fingers was recorded. The fencers performed free warm-up according to their training habits.

Using a dermatographic pencil, a line was drawn on the medial and lateral aspects of the humerus and forearm of the dominant arm, selected at the athlete’s discretion, which was then used to analyze the elbow angle in the surface EMG tests. With the participants in a sitting position, the sites of placement of electrodes were cleaned using cotton and alcohol. A disposable razor was used to shave the area. The area was cleaned with alcohol and dried using cotton wool for clean and dry skin. The positions of the electrodes were as follows: (1) clavicular deltoids, upper and anterior parts of the humerus, placed longitudinally, 4 cm below the acromion; (2) ADs, upper and lateral parts of the humerus, placed longitudinally, 4 cm below the acromion; (3) TB, medial and internal dorsal parts of the humerus, placed longitudinally; and (4) EL, first proximal, superior, and lateral thirds of the forearm. 

At each site, two electrodes were placed at a maximum distance of 2 cm from the muscle belly. A third electrode (grounded) was placed perpendicular to the electrode. The placement was unilateral over the dominant arm and the number of channels was one to four for the clavicular deltoid, AD, TB, and EL. A 30-mm Lessa Pediatric Electrode (Barcelona, Spain) was used. These were placed, as indicated by Seniam [30] and Criswell [31]. EMG recordings were obtained using a Megawin ME6000-T8 (Bittium Corporation, Oulu, Finland), which has eight channels and weighs approximately 344 g. The sampling frequency was 1 kHz and each sample lasted for 30 s. After recording, the EMG data were converted from analog to digital using the Biomonitor Megawin ME6000-T8 software (12-bit; DAQCard–700; National Instrument, Austin, TX, USA) and saved on a hard disk with a file extension of .ASC for protection and subsequent analysis. 

MATLAB (R2023a) (Mathworks Inc., Natick, MA, USA) was used for the signal analysis. First, a fourth-order Butterworth bandpass filter of 20–400 Hz is applied for signal filtering. Signal grinding or root mean square (RMS) analysis was performed by dividing the measurement section into 100 points. Finally, segmentation was performed by collecting the central and most stable 30 s signals. Smoothed data were collected using the smoothing data function. To determine the maximum force, two samples were recorded during a direct attack with the elbow at 170° flexion, trunk erect, and foil wielded in the direction of the target using a steel cable connected to a force sensor attached to a chain to a trellis and oriented in the direction of the attack (Chronojump, Barcelona, Spain) at a frequency of 160 Hz. Following the signal from the investigator, the fencer exerted as much force as possible for 5 s. Following a recovery time of 2 min to avoid fatigue, the action was repeated, and the best performance in Newtons (converted to kg) and the maximum value of muscle activation in microvolts on EMG were noted. 

For the speed of movement, the fencer was positioned at a distance of direct attack towards a plastron, selected by each fencer, moving his or her wheelchair until it was adjusted. The fencer performed a direct attack at the highest possible speed, without involving the trunk. Muscle activation was recorded in microvolts and each attempt was filmed to calculate the elbow angle from which the movement was initiated, thus establishing joint amplitude and speed (Kinovea v.0.95, Bordeaux, France). The test was performed three times with a rest period of 30 s between each test to avoid fatigue. Peak muscle activation and test duration with a shorter muscle time interval were noted, whereas the other two results were discarded. 

For fatigue, the fencer was positioned as in the previous test and 20 direct attacks were continuously performed at the fastest possible pace while returning to the guard position without using the trunk for movement (if not, it would be repeated after a pause of 2 min). Initially, a single attempt was performed and the activation peaks of each muscle, repetition duration, and order of activation were calculated to assess coordinative changes at the onset of fatigue (Figure 1). 

### 2.4. Statistical Analyses

All analyses were performed using the R4 v.4.2.2. (Auckland, New Zealand) [32]. The significance level was set at *p* = 0.05. Frequency and percentage statistics were used to calculate the prevalence. Non-parametric tests were applied without normality criteria, because not all characteristics were distributed normally. For activation changes during fatigue, normality was assumed following visual inspection according to the protocol of Zuur and Leno [33]. 

For the activation level in the direct attack, the Wilcoxon test and Pearson’s correlation analysis were used for the quantitative variables. To determine the order of muscle activation in a direct attack, a univariate analysis of discrepancies between the sequences and covariates was performed using the TraMineR package v.2.2-7 (Geneve, Switzerland) [34]. The dissimilarity matrix of muscle activation chains was determined to identify any significant univariate associations included in the multivariate model. 

To determine the duration at which the hit began to worsen, differences between the categories of athletes and sexes were determined using Wilcoxon test and Pearson’s correlation analysis with anthropometric variables. Using proportions, fatigue in each muscle was compared between the categories and sexes using the Fisher’s exact test. For the percentage of activation in each muscle, linear models of mixed effects were independently adjusted for each muscle using the lme4 v package 1.1-30 (Zurich, Switzerland) [35]. The overall fixed effects were analyzed using type III analysis of variance (ANOVA) and the adequacy of the model assumptions was assessed by visual inspection.

## 3. Results

Figure 2 illustrates the comparison between the force (kg) and time (ms) during a direct attack between the athletes in categories A and B. This trend indicated, without significant differences, that the strongest fencers had the highest mean value over time during a direct attack. Category B fencers had the lowest number of action lapses (W = 14, *p* < 0.1).

Table 2 summarizes the correlation analysis used to determine whether the quantitative variables are associated with the strength or speed of the best direct attack. The data revealed that only wingspan and sitting height were marginally correlated with strength (*p =* 0.072). This was especially evident in female athletes, for whom the Pearson’s correlation index surpassed 0.9.

The order of activation of the muscles involved in the direct attack revealed seven activation sequences. Overall, 42.8% of the athletes started with activation of the TB and another 42.8% began with activation of the EL. Univariate analysis revealed that sitting height-to-wingspan ratio and elbow angle correlated with different patterns of association (*p* < 0.05). Including these variables in the multivariate analysis revealed that only category (*F* = 2.075; *p* = 0.04) demonstrated an actual effect on the order of activation of the muscles involved in direct attack. Figure 3 illustrates two representative sequences in categories A (EL, TB, DA, and CD; TB, EL, CD, and DA) and B (DA, CD, EL, and TB; EL, TB, CD, and DA).

For fatigue, given the varying sequences found across the 20 hits, we focused on muscle activation sequences. Associations (*p* < 0.1) were observed in the sequences of the muscles that acted first and third among the four measured muscles. For the first muscle, an important association was found between the category (*p =* 0.05) and sitting height-to-wingspan ratio (*F* = 2.64; *p* = 0.02). Additionally, an association was noted between the third muscle and the sitting height (*F* = 1.78, *p* = 0.07). 

No significant differences were detected in the worsening point of the 20 hits between the two categories of athletes (W = 18; *p* = 0.24); however, Category B participants tended to reach the worsening point earlier (Figure 4). In contrast, differences were noted between the sexes (W = 23; *p* = 0.02); women reached the worsening point of the time lapse earlier than did men. Furthermore, only sitting height demonstrated a significant and positive correlation with worsening point, although this correlation was only evident in male athletes (r = 0.82; *p* = 0.04). Therefore, the higher the sitting height, the later the worsening point was reached.

## 4. Discussion

Strength training is key to improving speed in competitive sports and is an essential component of physical fitness. In high-level practices, the power of an action can determine victory. In this study, we found that the strongest fencers were not the fastest in executing the direct attack, which is a basic technique in fencing (Group A vs. B; strength: 49 ± 26.66 vs. 26.61 ± 10.59 kg, respectively; lapse: 162 ± 91.33 vs. 148.5 ± 59.51 ms, respectively). Improvement of the maximal force can result in a quick expression of movements (explosive and reactive) and, in conjunction with adequate technical work and control, can help predict the actual projection of the use of speed in competition [36]. 

A direct attack is the fastest attack and most commonly used in fencing competitions. Improving this maneuver is important, given that there are numerous occasions in which the distance from the opponent is short, and the speed of execution may mark the score, not only with the sword, where there is a double touch, but also with conventional blades, where priority is given to the attacker who initiates the action faster [37]. Both physiological and anthropometric aspects play a role in the execution of fastest attacks. Therefore, greater speed was noted in fencers with a greater sitting height and wingspan, especially in female athletes, for whom the Pearson correlation coefficient was 0.9, possibly because of their high mobility. At the technical level, ideal elbow flexion of 90° prior to extension [38] was confirmed by our results. These fencers used angles of 92–123° for the execution of the hit; the lower the value, the higher the speed (*p* = 0.014). Therefore, decreasing the guard angle facilitates a greater speed by promoting a greater acceleration trajectory. 

Additionally, knowledge of the muscles that act according to the athlete’s technical style is of great relevance. The level of mobility depends on the activation of agonist muscles, both in terms of the percentage and order of muscle activation. In Category A athletes, the most commonly used activation sequence included activation of the musculature near the grip of the weapon (TB, EL, CD, and DA), whereas in Category B athletes, the reverse was observed (DA, CD, EL, and TB). This finding supports the idea that designing exercises based on muscle participation is highly relevant in conditioning, because it allows adaptation to the characteristics of the athlete, and in this case, to each category. 

The order of muscle activation in Category B athletes, owing to greater restriction in their movements, indicates that the most activated muscle is the one that is most injured [2]. EMG helps to detect this type of unequal participation and correct it during training. However, the direct attack technique used in this study did not include trunk action, resulting in a different model of contraction than that proposed by Borysiuk et al. [2]. Although this study favored equal mobility conditions in its execution, including Category C fencers in other studies is possible. 

In addition to considering the technique in isolation, acyclic sports, especially combat sports, involve a high frequency of actions at a given time, which requires athletes to have highly efficient lactic acid metabolism to combat fatigue (4.2 ± 0.9 mmol/L in female fencers and 3.2 ± 0.7 mmol/L in male fencers) [39]. In fencing, this parameter is important because 15 touches are required in the absence of a technical “Knock Out” for victory. The number of technical actions within 3 min is very high and requires good energy availability to minimize technical errors, which are usually the result of incoordination due to fatigue. The duration of the direct attack was 162 ± 91.33 ms in Group A and 148.5 ± 59.51 ms in Group B athletes. During repeated actions, these actions may cease to be effective when the duration of the action is longer than the mean stop time (353 ± 28 ms) [19], thus forcing a change in tactics to prevent being touched. 

Almost all fencers maintained a high mean speed during the first few hits; however, during the 8th–12th hits in Groups A and 4th–10th hits in Group B, a drop in speed was observed without significant differences between the groups (*p* = 0.240). However, a significant difference was noted between sexes (*p* = 0.025), with women reaching the point of fatigue earlier. This finding indicates the need to propose a combat strategy with limited repetition during an attack. Subsequently, the tactics should be changed if the hits are unsuccessful to avoid a point for the rival before exceeding the average fatigue time. 

This turning point marks the athlete’s ability to maintain prolonged efficiency of hits and resistance to the explosive force. The involvement of muscles changes with each consecutive direct attack, altering both the percentage of muscle activation and order of participation. Category A athletes maintained greater homogeneity in the order of muscle activation, which agrees with the later onset of fatigue, whereas Category B athletes demonstrated frequent changes in the order of muscle contraction, which resulted in loss of coordination, especially with the accumulation of fatigue. Additionally, less fatigue was noted in individuals with shorter heights and greater wingspan. 

Detecting this action pattern is relevant in this competition, where less than 30 s (the minimum ATP return time) elapses between points [40]. This is crucial because repetitive movements must be executed at maximum speed throughout all rounds of a tournament. 

Training methods such as high-intensity interval training are highly beneficial in enhancing this parameter because they are based on performing exercises at maximum intensity with high density and reduced recovery time intervals owing to the intermittent nature of this practice [41].

## 5. Conclusions

Based on our results, maximal force is a good indicator of the final speed that the athlete may apply in a direct attack, and fencers with a greater wingspan and sitting height have a greater advantage. The highest efficiency was observed for direct attacks performed at different angles, preferably when the elbow was closed. The activation pattern in the direct attack differs between categories of athletes; although the maximum speed does not change significantly between the categories, preventive conditioning of the muscles of the shoulder girdle should be considered, especially for Category B athletes.

Fatigue was usually evident on the sixth hit. Therefore, adopting different tactics when the number of strokes exceeds this threshold during a competition could be an interesting conditioning strategy that focuses mainly on high-intensity interval training. Designing specific conditional training for everyone is important, considering that fatigue in this sport is progressive and maintaining good coordination during successive attacks is fundamental. The ad hoc protocols used in this study provided more information on muscle behavior with and without fatigue intervention, potentially contributing to improvements in the WCF training.

The main limitations are the small sample size, as it is a sport with low participation, and the difficulty of recruiting elite fencers. On the other hand, the possibility of performing bilateral measurements, including trunk musculature measurements, opens a new line for future work.

## Figures and Tables

**Figure 1 sports-12-00188-f001:**
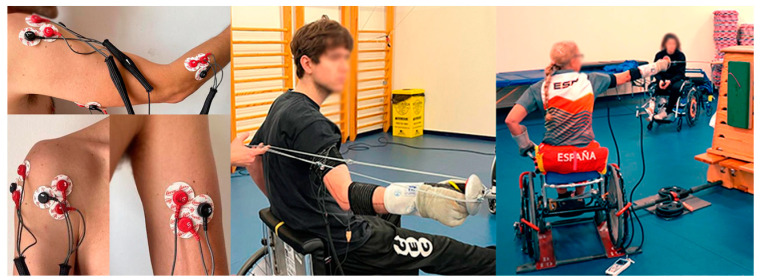
Electrode placement (**left**), test procedure for maximal isometric force (**center**), direct attack, and fatigue attack (**right**).

**Figure 2 sports-12-00188-f002:**
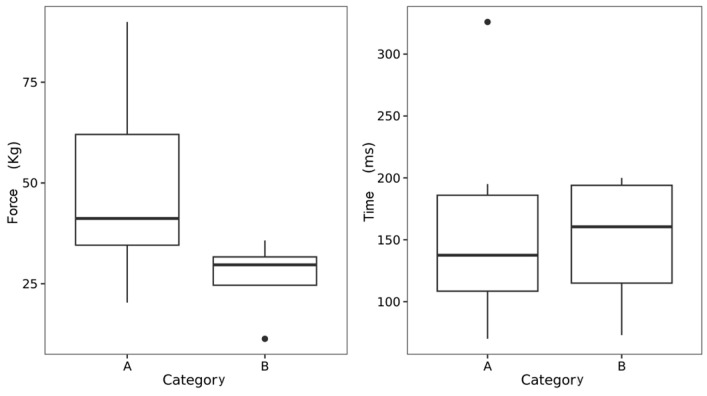
Differences in strength and lapse depending on the category. Force, lapse; category. The minimum and maximum values in force and time respectively are represented by a black dot.

**Figure 3 sports-12-00188-f003:**
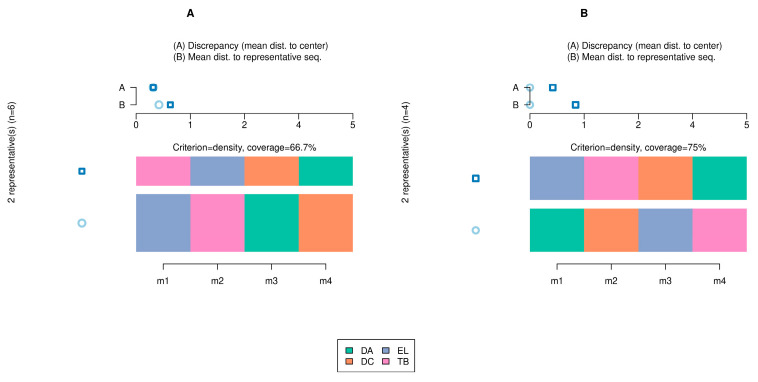
Order of muscle activation in the best direct attack according to the category. (**A**): Category A; (**B**): Category B. Representative activation sequences for both categories (colored muscles) and discrepancies (blue circles) between them are shown.

**Figure 4 sports-12-00188-f004:**
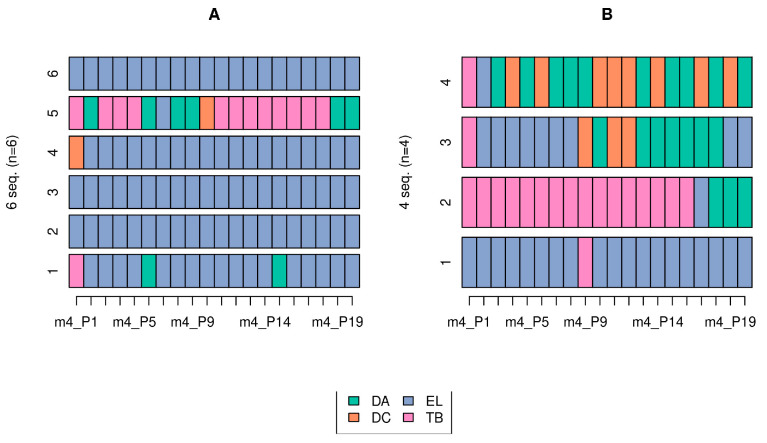
Order of muscle activation in 20 direct attack hits according to category. (**A**): Category A; (**B**): Category B. The order of muscle participation in both categories (colored muscles) in the 19 measured time intervals (P1–P19).

**Table 1 sports-12-00188-t001:** Descriptive anthropometric and technical data.

Variable			Global	Category A	Category B
N			10	6	4
Age (years)		x SD ±	34.7 ± 15.22	39.67 ± 16.22	27.25 ± 11.53
Sex	Male	n (%)	6 (60%)	4 (66.67%)	2 (50%)
	Female	n (%)	4 (40%)	2 (33.33%)	2 (50%)
Sitting height (m)		x SD ±	0.87 ± 0.09	0.92 ± 0.05	0.8 ± 0.08
Wingspan (m)		x SD ±	1.76 ± 0.14	1.81 ± 0.06	1.69 ± 0.21
Experience (years)		x SD ±	7.8 ± 5.14	9 ± 6.1	6 ± 3.16
Laterality	Right handed Left handed	n (%) n (%)	9 (90%) 1 (10%)	5 (83.33%) 1 (16.67%)	4 (100%) 0 (0%)
Elbow angle		Degrees (º)		103.67 ± 10.03	112 ± 11.46
Hit lapse		Times (ms)		162 ± 91.33	148.5 ± 59.51

**Table 2 sports-12-00188-t002:** Pearson correlation analysis of quantitative variables, strength, and speed of the best movement.

Variable	Fencers	Force R	*p*-Value	Lapse R	*p*-Value
Sex	All	0.554	0.097	−0.221	0.539
	Male	0.297	0.568	−0.231	0.659
	Female	0.923	0.077	−0.645	0.355
Sitting height (m)	All	0.596	0.069	−0.033	0.928
	Male	0.371	0.469	−0.03	0.955
	Female	0.928	0.072	−0.434	0.566
Elbow angle	All	−0.337	0.341	−0.513	0.13
	Male	0.662	0.152	−0.543	0.266
	Female	0.591	0.409	−0.803	0.197

## Data Availability

The datasets used and analyzed during the current study are available from the corresponding author upon reasonable request due to privacy and ethical restrictions.

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
