# Peer review of "Muscle Changes during Direct Attack under Different Conditions in Elite Wheelchair Fencing"

_sports, 2024, doi:10.3390/sports12070188_

Round 1

Reviewer 1 Report

Comments and Suggestions for Authors

Dear Authors

You have written an interesting paper focusing on the analysis of muscle changes to determine individual time thresholds and improve athletes' conditioning.

However, some parts need to be addressed for greater clarity.

In the abstract, the main aim is not present. Also, in the introduction, the aim is not clearly defined - please redefine this paragraph.

Methods:

Delete the word patients from your manuscript - line 130!

Add- written consent.

At what time of day did you perform the measurements? report

How did you perform the anthropometric measurements and with what equipment - was the assessor ISAK accredited? Be specific!

So the sitting height was subtracted from the sitting height to get the actual height? Really??? This does not make any sense - add a reference and correct the description.

Report how did you determine the dominant arm.

Add a photo or a drawing of the electrode's position for greater reproducibility.

Was there any warm-up before the test?

How was the force sensor fitted? In the picture, you are holding it? si this correct?

Describe the movement for fatigue conditions precisely - was the distance to the target predetermined or it was self-selected, did they just touch the target mark - what was the instruction and what happened if they didn't - did they do 1 more- who counted it - in fencing, you need some power for the sensor to register a git - that is why I ask and it makes a difference for practical application of the test.

Where is the limitations of the study paragraph - add it

Overall the study is solid, however, it still needs some further work.

Kind regards

Comments on the Quality of English Language

Moderate editing of the English language required

Author Response

  • In the abstract, the main aim is not present. Also, in the introduction, the aim is not clearly defined - please redefine this paragraph.

Accordingly, in the summary on the first page, the objective of the research has been included.

  • Delete the word patients from your manuscript - line 130!

Done, replaced by participants (line 136)

  • Add- written consent.

All right, we attach it (to editor, here we are not allowed to attach more than one file.)

  • At what time of day did you perform the measurements? report

We have included it in the paper (lines 121 and 122).

  • How did you perform the anthropometric measurements and with what equipment - was the assessor ISAK accredited? Be specific!

We have specified this (lines 148 and 149)

  • So the sitting height was subtracted from the sitting height to get the actual height? Really??? This does not make any sense - add a reference and correct the description

Obviously it was not well expressed, we have changed the description, for better understanding (lines 150-154)

  • Report how did you determine the dominant arm

the method of selection has been indicated (line 160)

  • Add a photo or a drawing of the electrode's position for greater reproducibility.

Figure 1 has been modified by adding the electrode placement position (lines 210-211).

  • Was there any warm-up before the test?

This aspect has been pointed out (lines 157-158)

  • How was the force sensor fitted? In the picture, you are holding it? si this correct?

How to fix it has been described (lines 188-189)

  • Describe the movement for fatigue conditions precisely - was the distance to the target predetermined or it was self-selected, did they just touch the target mark - what was the instruction and what happened if they didn't - did they do 1 more- who counted it - in fencing, you need some power for the sensor to register a git - that is why I ask and it makes a difference for practical application of the test.

We agree with the issues, the distance has been clarified in lines 195 and 196. In case of not respecting the test conditions, it has been clarified how to proceed in lines 205 and 206.

  • Where is the limitations of the study paragraph - add it

Limitations have been added at the end of the text (lines 370-373).

In addition, efforts have been made to improve the language of the text.

We hope you are satisfied with the proposed corrections

Reviewer 2 Report

Comments and Suggestions for Authors

I would like to appreciate the efforts of the authors in implementing the project and writing this article "Muscle changes during direct attack under different conditions in elite wheelchair fencing."

This paper aims to estimate maximal and dynamic isometric muscle activation along with the timing of a direct attack, considering the shoulder and arm muscles, which would allow the creation of a comparative model of various categories of athletes in this sport.

This article provides interesting information about the influence of maximum force on final speed in elite wheelchair fencing and the influence of fatigue on selected performance parameters. It is very difficult to get a large number of relevant participants on a given topic, but I have some comments and questions:

I recommend a more precise formulation of the aim of the thesis in connection with this, a more concise formulation of the conclusion.

Were all participants in each category comparable in terms of disability? There is already a very small number of participants (4 and 6), moreover some characteristics were not distributed normally. Can the conclusions from such research be taken as relevant and generalizable? I believe that inter-individual differences in disability within one category (if any) will play a not inconsiderable role.

Is it appropriate to use the Pearson correlation coefficient in this case if it is stated that the data is not distributed normally?

Author Response

  • I recommend a more precise formulation of the aim of the thesis in connection with this, a more concise formulation of the conclusion.

Thank you for your comment, we have added the objective of the study in the summary and also added a section on limitations at the end of the conclusions.

  • Were all participants in each category comparable in terms of disability? There is already a very small number of participants (4 and 6), moreover some characteristics were not distributed normally. Can the conclusions from such research be taken as relevant and generalizable? I believe that inter-individual differences in disability within one category (if any) will play a not inconsiderable role.

Thank you for your comment. With all due respect, we would like to point out that as this is the main shortcoming of the other articles, as we indicated in the introduction, in this work we have dispensed with the use of the trunk in the action in order to be able to compare the possibilities of category A and B, and to be able to establish this comparison, in this case.
It is true that the number of participants is small, but the Spanish team is present in full, with the added difficulty of recruiting other international elite level shooters. We think that in spite of these details it is representative for this sport. We have added this aspect to the limitations of the conclusions, and of course it opens a new avenue for research.

  • Is it appropriate to use the Pearson correlation coefficient in this case if it is stated that the data is not distributed normally?

Thank you, with all due respect, we think so. It has been used after applying the Wilcoxon test, and, having determined that there were linear associations between variables, we have decided to use it since it is one of the conditions that must be met.

In addition to these comments, we have made a number of improvements which I hope will be of interest to you (marked in red) and which we hope will contribute to further improve your opinion of the work.
Thank you very much for your effort and comments

Round 2

Reviewer 1 Report

Comments and Suggestions for Authors

Dear Authors,

Thank you for addressing all of my comments and suggestions. Overall in my opinion the quality and reproducibility of the manuscript improved to a point that it is acceptable for publication.

Kind regards

Reviewer 2 Report

Comments and Suggestions for Authors

My questions have been answered and my comments resolved. I have no further comment.